# Urtica dioica Agglutinin: A plant protein candidate for inhibition of SARS-COV-2 receptor-binding domain for control of Covid19 Infection

**Fatemeh Sabzian-Molaei[1,2], Mohammad Ali Nasiri Khalili[1]\*, Mohammad Sabzian-Molaei[3], Hosein Shahsavarani[4,5], Alireza Fattah Pour[5,6], Ahmad Molaei Rad[1], Amin Hadi[7]**

**1** Faculty of Chemistry and Chemical Engineering, Malek Ashtar University of Technology, Tehran, Iran, **2** Department of Biology, Faculty of Science, Bu-Ali Sina, University, Hamedan, Iran, **3** Social Determinants of Health Research Center, Lorestan University of Medical Sciences, Khorramabad, Iran, **4** Department of Cell and Molecular Sciences, Faculty of Life science and Biotechnology, Shahid Beheshti University, Tehran, Iran, **5** Laboratory of Regenerative Medicine and Biomedical Innovations, Pasteur Institute of Iran, Tehran, Iran, **6** Department of Agricultural Biotechnology, Faculty of Agriculture Science, University of Guilan, Rasht, Iran, **7** Cellular and Molecular Research Center, Yasuj University of Medical Sciences, Yasuj, Iran

\* manasiri@mut.ac.ir, manasiri@alumni.ut.ac.ir

**Data Availability Statement:** All relevant data are within the paper and its Supporting Information files.

## Abstract

Despite using effective drugs and vaccines for Covid 19, due to some limitations of current strategies and the high rate of coronavirus mutation, the development of medicines with effective inhibitory activity against this infection is essential. The SARS-CoV-2 enters the cell by attaching its receptor-binding domain (RBD) of Spike to angiotensin-converting enzyme-2 (ACE2). According to previous studies, the natural peptide Urtica dioica agglutinin (UDA) exhibited an antiviral effect on SARS-CoV, but its mechanism has not precisely been elucidated. Here, we studied the interaction between UDA and RBD of Spike protein of SARS-CoV-2. So, protein-protein docking of RBD-UDA was performed using Cluspro 2.0. To further confirm the stability of the complex, the RBD-UDA docked complex with higher binding affinity was studied using Molecular Dynamic simulation (via Gromacs 2020.2), and MM-PBSA calculated the binding free energy of the system. In addition, ELISA assay was used to examine the binding of UDA with RBD protein. Results were compared to ELISA of RBD-bound samples of convalescent serum IgG (from donors who recovered from Covid 19). Finally, the toxicity of UDA is assessed by using MTT assay. The docking results show UDA binds to the RBD binding site. MD simulation illustrates the UDA-RBD complex is stable during 100 ns of simulation, and the average binding energy was calculated to be -47.505 kJ/mol. ELISA and, MTT results show that UDA binds to RBD like IgG-RBD binding and may be safe in human cells. Data presented here indicate UDA interaction with S-protein inhibits the binding sites of RBD, it can prevent the virus from attaching to ACE2 and entering the host cell.

**Funding:** The author(s) received no specific funding for this work.

**Competing interests:** The authors have declared that no competing interests exist.

## Introduction

Over the past two decades, coronaviruses, including SARS-CoV (2002) and MERS-CoV (2012), have caused acute respiratory syndromes [1]. A recent case of the new human coronavirus, "SARS-CoV-2", which originated in Wuhan, China, in December 2019, caused a severe outbreak of pneumonia that has infected millions of people worldwide [2]. On January 30, 2020, the World Health Organization declared the outbreak of Covid-19 disease an international public health emergency concern, and then in March 2020, the organization confirmed that the condition was pandemic [3]. Covid-19 deaths are currently much higher than those of SARS or Mers coronaviruses, so this disease became one of the most significant epidemics in human history [4]. SARS-CoV-2 is an RNA virus belonging to the Coronaviridae family [5]. The spike (S) protein, nucleocapsid (N) protein, and envelope (E) protein are the main protein structures of the virus. The Spike is a protein needed to infect host cells and is encoded by the virus genome [6]. The glycoprotein spike is composed of three domains that bind to the angiotensin-converting enzyme 2 (ACE2) through the outer domain, s1 subunit, receptor-binding domain (RBD) [7]. ACE2 is a transmembrane zinc protein mainly expressed in the lung's epithelial cells and mediates the fusion of cell and viral membranes and opens the door for the virus genome into host cells [8] (Fig 1). Therefore, inhibiting the binding of the virus to ACE2 is one of the targets of treatment. Although different drugs and vaccines have been prescribed and produced to prevent and treat this human-to-human transmission, some have serious side effects [9]. Moreover, SARS-CoV-2 is an RNA virus with a high mutation rate so that novel SARS-CoV-2 related viruses may appear, just as SARS-CoV-2 emerged after SARS-CoV [10]. New SARS-CoV-2 mutants, such as Omicron and Delta, may be safe from the effects of current vaccines, and this is a significant challenge for current treatment [11]. All SARS-CoV-2 related have entered the host cell through ACE2, so targeting ACE2 and RBD binding as therapeutic target is more widely effective against coronaviruses [12, 13]. RBD residues that interact with ACE2 include Phe490, Tyr505, Gly502, Gly496, Asn501, Thr500, Gln498, Tyr449, Gly446, Gly476, ALAa75, Asn487, Phe486, Tyr489, Phe456, Lys417, Gln493, Leu455, and Tyr453 (Table 1).

After the outbreak of SARS-CoV, in an in vitro study, Inhibition of SARS-CoV infection was confirmed by UDA in a lethal model of BALB / c mouse [14]. The World Health Organization suggested Urtica dioica agglutinin (UDA) as an antiviral against SARS-CoV-2. Generally, natural resources can be a safe alternative to some medicines [15]. Siting Nettle lectin, UDA is a small plant monomer peptide (molecular weight 8.7 kDa and 86 amino acids) with the property of binding to N-acetyl glucose amine. This peptide is very stable, so its activity is retained at pH 1.0 and 80˚C for up to 15 minutes [16]. Recently an in silico study also examined the binding properties of UDA to N-linked glycans of SARS-CoV-2 spike glycoprotein [17]. However, the main and definitive reasons for inhibiting the infection by UDA have not yet been identified. In this article, for the first time, we decided to examine the interaction between UDA and RBD of S-protein of SARS-CoV-2. Thus, protein-protein docking of RBD-UDA was performed. To further confirm complex stability, the best-docked complex was subjected to Molecular Dynamic simulation through Gromacs. The binding free energy of the system was calculated by Molecular mechanics Poisson–Boltzmann surface area (MM-PBSA). To confirm in silico studies, the binding of RBD and ACE2 by Enzyme-linked immunosorbent assay (ELISA) was also investigated. Finally, the toxicity of UDA was assessed by MTT. The results indicate that the binding of UDA to the RBD of S-protein inhibits the RBD of S-protein and ACE2 interaction, so it can prevent the virus from entering the host cell. In this way, UDA can be a potential candidate drug against SARS-CoV-2. However, additional experimental studies are needed to confirm this capability.

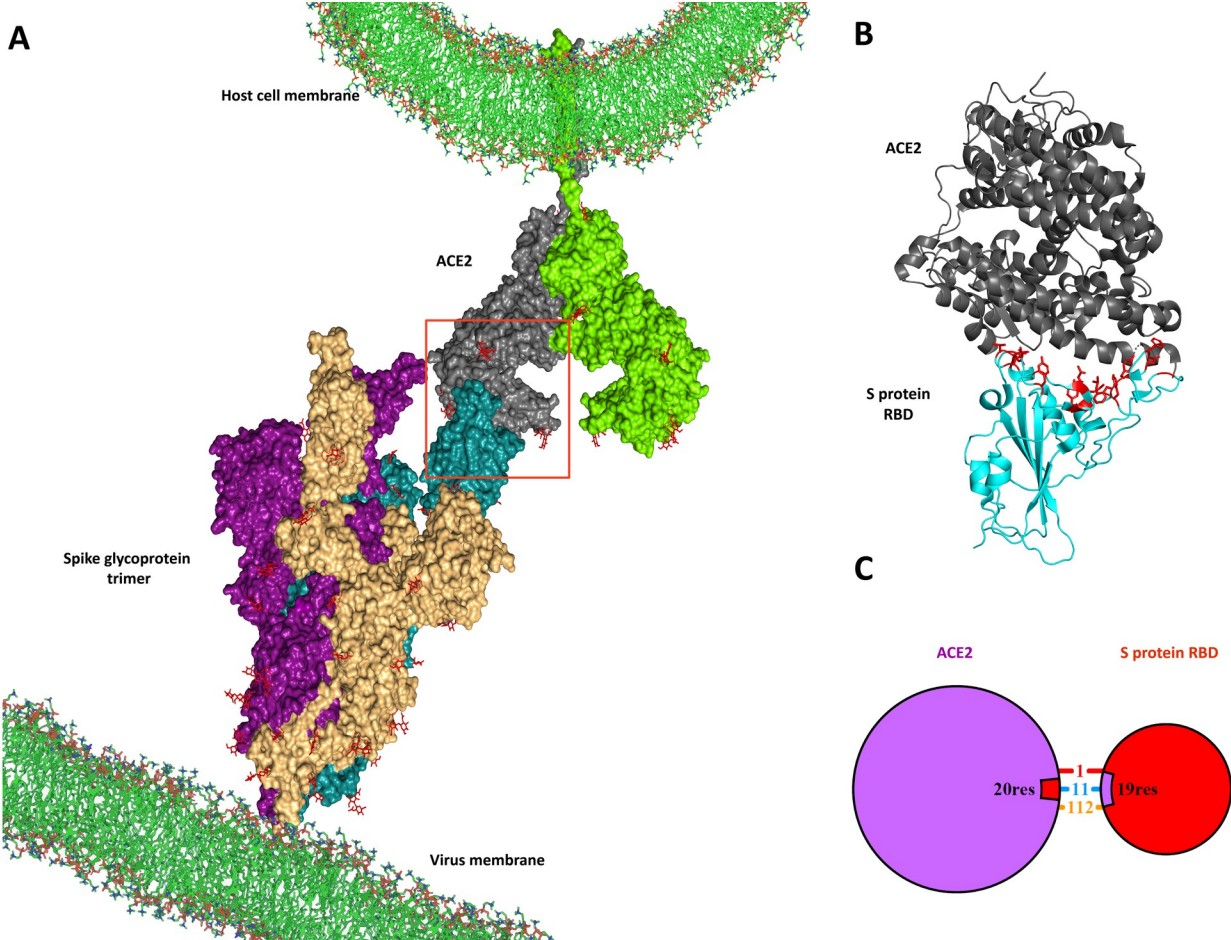

**Fig 1.  a.** Attachment of SARS-CoV-2 to host cell **b.** The crystal structure of the RBD-ACE2 complex, RBD residues that interact with ACE2, are shown as red sticks (PDB ID: 6lzg). **c.** RBD-ACE2 Interactions (salt bridge: red, Hydrogen bond: blue, non-bonded contact: orange).

**Table 1.  List of interacting residues between RBD and ACE2.**

| interactions | RBD | ACE2 | Distance (A˚) |
|---|---|---|---|
| | ALA 475 | SER 19 | 2.81 |
| | ASN 487 | GLN 24 | 2.87 |
| | LYS 417 | ASP 30 | 2.86 |
| | TYR 449 | ASP 38 | 2.83 |
| | Tyr 449 | GLN 42 | 3.09 |
| Hydrogen Bond | THR 500 | TYR 41 | 2.73 |
| | GLN 498 | GLN 42 | 2.57 |
| | ASN 487 | TYR 83 | 2.82 |
| | GLY 502 | LYS 353 | 2.79 |
| | LYS 353 | GLY 496 | 3.22 |
| | TYR 505 | GLU 37 | 2.08 |
| Salt Bridge | LYS 417 | ASP 30 | 2.86 |

# Materials and methods

## Preparation of chemicals and reagents

The recombinant receptor-binding domain (RBD) of SARS-CoV-2 was supplied by Pasteur Institute (Tehran, Iran). The human embryonic kidney cell line (HEK 293) was supplied by the cell bank of Pasteur Institute (Tehran, Iran).

**Extraction and Purification of UDA.** Rhizomes of stinging nettle were collected during the winter. UDA was purified by the previously described method (Based on affinity chromatography on chitin and ion-exchange chromatography on SP-Sephadex) (16).

**Purity analysis by RP-HPLC and SDS-PAGE.** UDA purity analysis was evaluated using High-Performance Liquid Chromatography on the Reverse Phase column. RP-HPLC was performed using Capital C18 BDS Hypersill column (4.6 x 250 mm, 5μm) on KNAUER system (pump-K1001, Auto sampler 3800, and UVdetector K-2600). The column was equilibrated with mobile phase A (Water, 0.1% TFA), and the flow rate was 1 ml/min. Elution of Peptide was performed with a linear gradient from mobile phase A (Water, 0.1% TFA) to mobile phase B (ACN, 0.1% TFA). The total time for the method was 40 min, and the elution of the column was monitored at 214 nm. In addition, the resulting UDA was determined by SDS-PAGE (15%) Coomassie Brilliant Blue staining [18].

## Preparation of proteins

The crystal structure of SARS-CoV-2 S-protein RBD and UDA were obtained in PDB format from the Protein Data repository (https://www.rcsb.org/) repository: ID 6lzg (resolution 2.5 Å) and ID 1enm (resolution 1.9 Å)

## Molecular docking for UDA and SARS-CoV-2 S-protein RBD

Molecular docking for UDA and RBD was done using the ClusPro 2.0 webserver to investigate how these two proteins interact. An auto docking tool, ClusPro 2.0, was utilized to investigate the interactions of the UDA protein and RBD of S protein (6lzg: chain B). This tool can help screen docked complex and their cluster belongings by considering divergent parameters of proteins [19]. ClusPro is available at https://cluspro.bu.edu/publications. php. The docking modes generated from Cluspro were visualized in PyMOL [20]. The selection of the best RBD-UDA complex was based on two factors: The first factor was the lowest energy using the balanced method based on the PIPER docking algorithm, which is calculated based on the following expression the interaction energy between two proteins: $E = w_1 E_{rep} + w_2 E_{attr} + w_3 E_{elec} + w_4 E_{DARS}$, here Erep and Eattr show the repulsive and attractive contributions to the van der Waals interaction energy. Eelec is the electrostatics energy and EDARS is a pairwise structure-based potential constructed by the Decoys as the Reference State (DARS) approach. Coefficients within the equation w1, w2, w3, and w4 define the weights of the terms [19, 21]. The second choosing factor was more attachment to the active site of RBD. Finally, The best complex was selected and analyzed using PDBSUM and LigPlot+ [22]. The binding free energy of the best-docked model was calculated in PRODIGY, an online protein binding energy prediction server for ΔG affinity binding (kcal/mol) and complex stability. PRODIGY is a set of web-based services available at (https://bianca.science.uu.nl/prodigy/). PRODIGY focuses on predicting binding affinity within biologic complexes [23]. The best-docked structure (PDB format) is presented in the S1 File.

## Molecular dynamics

Molecular dynamic simulation of RBD of SARS-CoV-2 with UDA has been performed to analyze complex stability and interactions between the structural aspects and functional relevance. Molecular dynamics for 100 ns were done using the GROMACS 2020.2 software program: SPC216 water model in a cubic box and amber99sb.ildn force field. Adding a suitable number of $CL^-$/$Na^+$ ions neutralized the system. Therefore, for free RBD, RBD-UDA, RBD-ACE2, and Free UDA simulations, $2CL^-$, $3CL^-$, $24Na^+$, and $1CL^-$ were added, respectively. After adjusting the temperature at 300 K (NVT = 1ns) and the pressure at 1 bar (NPT = 1ns), the simulations were run for 100 ns for the RBD of S protein, Free UDA, RBD-ACE2, and RBD-UDA complex. Root mean square deviation (RMSD) of the backbone atoms of structures. The radius of gyration (Rg), Solvent-accessible surface area (SASA), Root mean square fluctuation (RMSF) of the α-carbons of systems during the simulations, intermolecular hydrogen bond profile of active site of complexes, distance fluctuation plot (mindist), and comparison of residual contact patterns between UDA peptide and RBD of S protein at 0 ns and 100 ns were studied, contact maps were visualized using CMWeb [24].

## Calculating free binding energy using the MM-PBSA method

MMPBSA is an abbreviation for Molecular Mechanics Poisson–Boltzmann Surface Area method; this method was used to calculate the free binding energy [25]. It was done in the g_mmpbsa tool for every one ns of 100 ns simulation. UDA was considered a ligand in the RBD-UDA complex.

## Enzyme-linked immunosorbent assay (ELISA)

In brief, the microwells of ELISA were coated with Serial dilutions from 1:2 to 1:16 prepared of UDA were mixed with PBS to a total volume of 100 μl at 4°C overnight. Then the wells were blocked with 0.3% skim milk for one h at room temperature. Then, 0.5 μg.mL recombinant RBD were added into each well at a volume of 100 μl/well and incubated for one h at RT. After washing three times with PBST (5 min each), Rabbit anti-his antibodies were added (diluted 1:2000), and incubation was done for 1 hour at RT. The wells were re-washed with PBST and secondary antibody (HRP-conjugated goat ant rabbit at 1:5000 dilution) and were incubated for one h at RT. Following washing, for color development, 100 μl of TMB substrate was added to wells and incubated at RT for 10-15min in darkness. The reaction was stopped by adding 2N sulfuric acid. Finally, the absorbance was measured at 450 nm by BioTek ELISA reader. (Wells without UDA were set as the negative control, and wells treated with human convalescent serum IgG instead of UDA were set as a positive control.)

## MTT assay

MTT assay Cell viability was examined by 3-(4, 5-dimethylthiazol-2-yl) - 2, 5-diphenyltetrazolium bromide (MTT) assay [26]. First, $10^4$ HEK293 cells in 100 μl of fresh serum IgG DMEM were seeded in 96-well plates and incubated overnight at 37°C under 5% CO2 in an incubator. Serial dilutions 1:3 prepared of UDA stock in a volume of 100 microliters, added to the cells the next day, and incubated for 48 hours under cell growth conditions (37°C and 25% CO2). After 48 hours, the Cell culture medium was removed entirely, and MTT (5 mg/ml) was added to each well and incubated for four h at 37°C in the dark. Finally, the MTT medium was removed, and 80 μl of dimethyl sulfoxide (DMSO) was added to each well. After shaking for 20 min in the dark, the optical density (OD) was detected at 570 nm with a Microplate Reader.

### Statistical analysis

The experiment is performed with three replications. The mean standard deviation is used to represent data. The statistical analysis was conducted by one-way analysis of variance (ANOVA) using SPSS software. Statistical significance was accepted as $P < 0.05$.

## Results

### Docking of UDA and RBD of S-protein of SARS-CoV-2

UDA docked at the RBD of spike protein is shown in Fig 2B, which in comparison with Fig 2A (crystal structure of the RBD-ACE2 complex), illustrates the similarity in the orientation of UDA and ACE2 on the surface of RBD. The residues contacts of complexes, RBD-UDA, and RBD-ACE2 interactions are also shown in Fig 2C, the residues Tyr505, Gly502, Gly496, Asn501, Thr500, Gln498, Tyr449, Gly446 Tyr489, Phe456, Lys417, Gln493, Leu455, and Tyr453 in RBD are the same in connection to UDA and ACE2. The Ligplot[+] also confirms this (Fig 3). The binding energies of RBD-UDA and RBD-ACE2 complexes were calculated at -12.2 kcal/mol and -12.4 kcal/mol, respectively. These two numbers are close to each other (Table 2).

### Molecular dynamic simulations

Molecular dynamics simulations have been performed to evaluate the stability of RBD-UDA. It was found that when free RBD was simulated, the RMSD of the backbone fluctuated widely. However, at the time of simulation of RBD in the RBD-UDA and RBD-ACE2 complexes, there were fewer fluctuations in the RMSD for 100 ns. Also, the RMSD value for RBD in RBD-UDA is lower than the RBD in the RBD-ACE2 complex (Fig 4A); it indicates that the RBD-UDA complex is more stable. Thus, UDA binding may inhibit the RBD of S protein. The RMSD values for the free UDA and UDA in the UDA-RBD complex were also approximately equal (Fig 4B).

The Rg and SASA plots for RBD in the RBD-UDA and RBD-ACE2 complexes show stable fluctuations over the time of the simulations (Fig 5). The slight decrease in the Rg and SASA

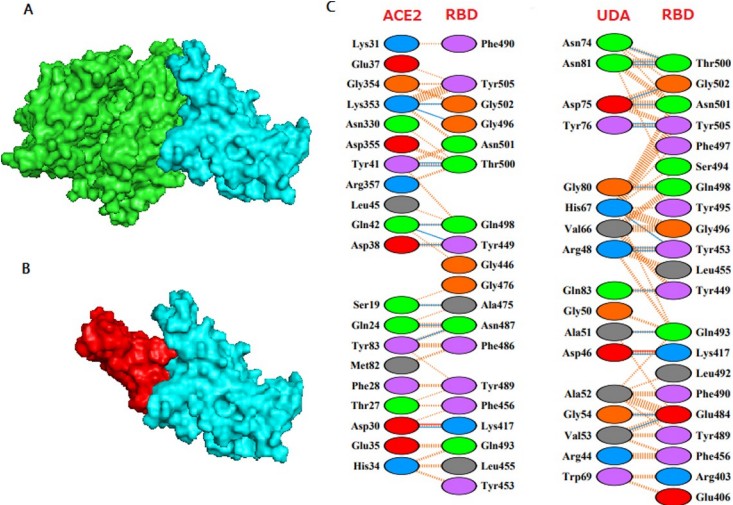

**Fig 2. a** RBD-ACE2 X-ray crystallography (PDB ID: 6glz).ACE2 is shown green, and RBD is shown cyan. **b** UDA–RBD docked complex. RBD is shown cyan, and UDA is shown red. **c** Analyzing various interactions in RBD-ACE2 and RBD-UDA using PDBsum [27].

**A**

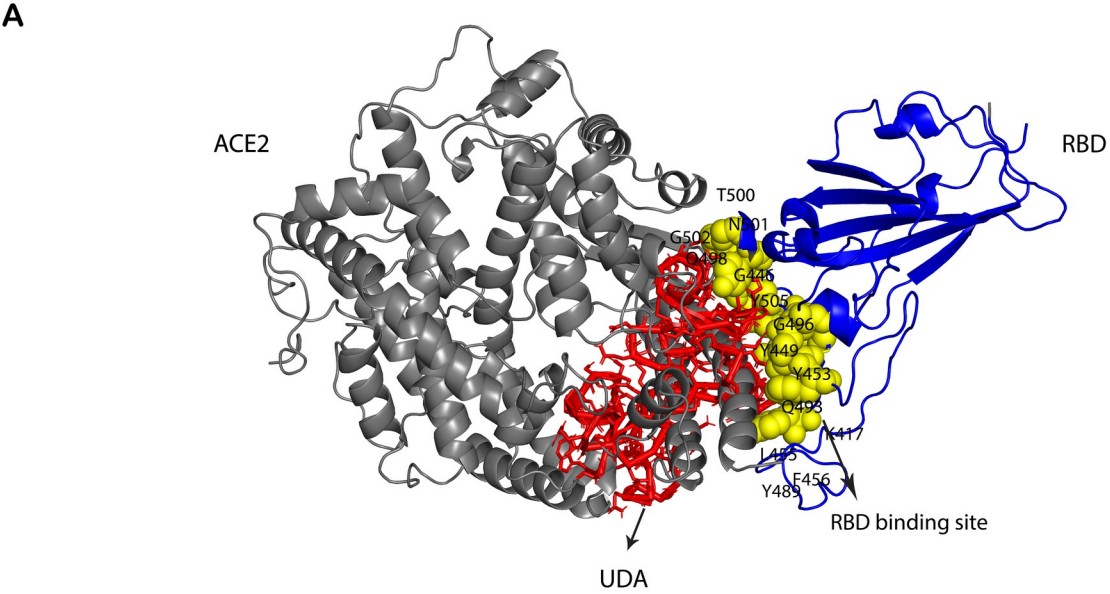

**B**

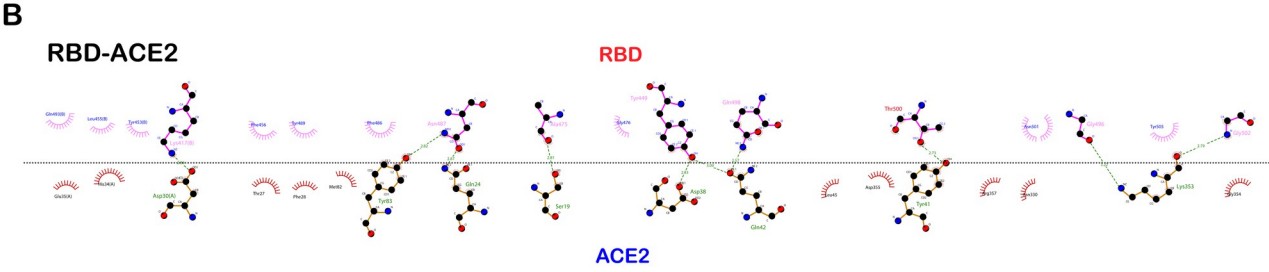

**C**

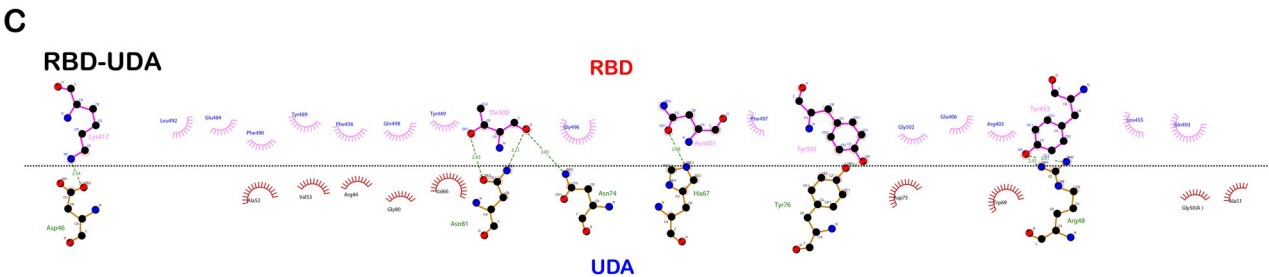

**Fig 3.** Comparison of the UDA-RBD docked complex with the crystal structure of RBD-ACE2 (a). LigPlot[+] diagrams of RBD-ACE2 crystal structure (b) and docked RBD-UDA interactions (c).

values of RBD in the RBD-UDA complex indicates that binding to UDA may increase the overall compactness of RBD.

RMSF analysis shows the average fluctuations of a group such as alpha carbon during the simulation by the g-rmsf program. The graph of this program identifies the flexible and rigid points of the protein. Groups that fluctuate the most are the flexible points of the protein. It was found that RMSF plots of UDA-bound RBD and ACE2-bound RBD are significantly less flexible than the free RBD plot (Fig 6).

**Table 2. Comparing the various types of interactions between RBD-UDA and RBD-ACE2 (PDB ID: 6glz).**

| Type of contact | ACE2–RBD complex | UDA–RBD complex |
|---|---|---|
| Charge–charge | 3 | 3 |
| Charge–polar | 10 | 24 |
| Charge–nonpolar | 19 | 24 |
| Polar–polar | 5 | 6 |
| Polar–nonpolar | 23 | 20 |
| Nonpolar–nonpolar | 9 | 23 |

The number of H-bonds between active sites of RBD and ACE-2 and UDA, calculated using the default settings of the g-h bond tool in Gromacs. In Fig 7A and 7B, it is observed that UDA is making 4–18 hydrogen bonds with the active site residues of RBD during 100 ns simulations; compared to the H bonds between the active site of RBD and ACE2 (2–16), it has been shown that the UDA establishes more hydrogen bonds during simulation with the active site of RBD.

On the other hand, the Occupancy of the hydrogen bonds between UDA and ACE during the simulation shows in the Table 3, which confirms that the UDA maintains its hydrogen bonds with active site residues of RBD such as TYR505, GLY502, ASN501, THR500, GLN498, GLY496, PHE490, TYR453 and TYR449 during simulation.

The distance between centers of active sites of RBD and UDA and centers of active sites of RBD and ACE2 was analyzed using GROMACS software, and the results showed that UDA does not separate from RBD like ACE2. The snapshots were taken at different simulation times to confirm their stable connection (Fig 7C).

The variation of the residual contact before the 100ns simulation and after that is illustrated in Fig 8, and the results are shown that most of the connected residues have maintained their connection at the end of the simulation.

To clarify the energetics of the binding of UDA with RBD, The binding free energy is calculated by MM-PBSA. Results showed average binding energy of -47.505kJ/mol during the 100ns of simulation. The various energy parameters were also calculated and shown in Table 4.

A

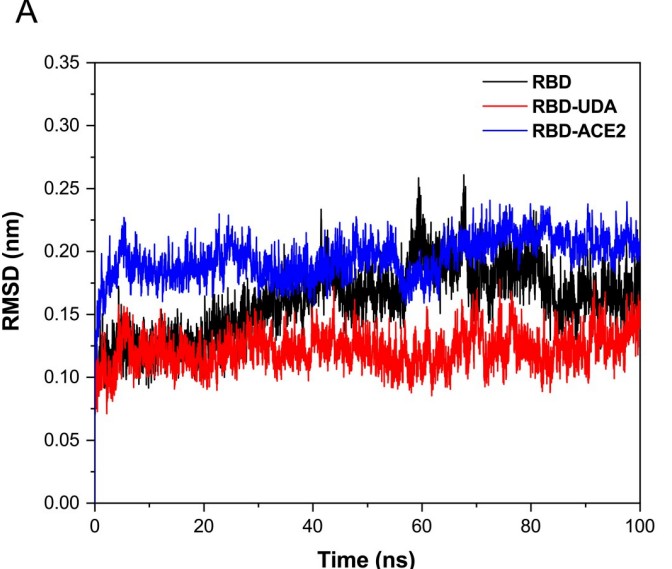

B

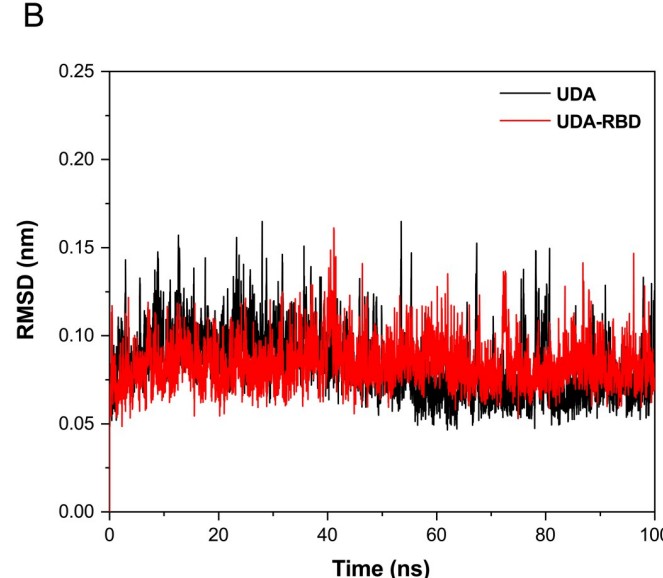

**Fig 4.** RMSD (backbone) plots for free RBD and RBD in the RBD-UDA, and RBD-ACE2 complexes (a). RMSD (backbone) of free UDA and UDA in the RBD-UDA complex during 100 ns of molecular dynamics simulation (b).

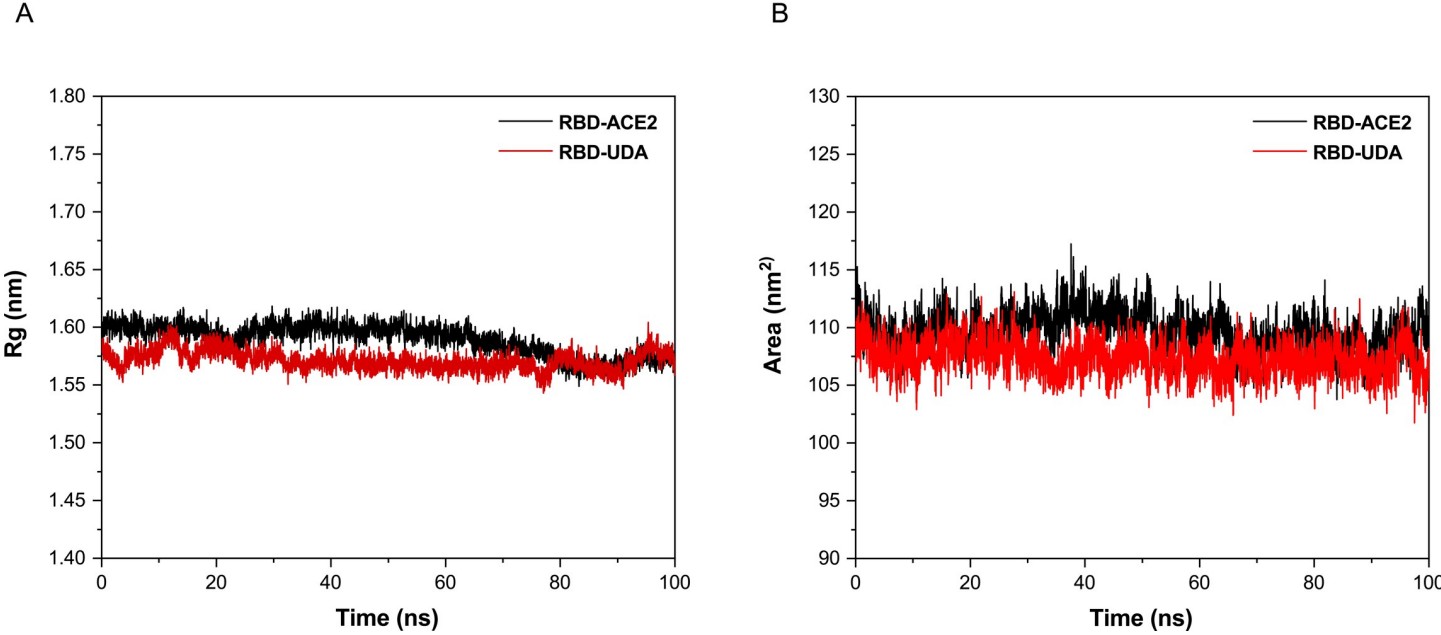

**Fig 5.** Plots of Rg (a) and SASA (b) for RBD in RBD-UDA and RBD-ACE2 complexes.

## Purification analysis of UDA

The purity of UDA was evaluated using HPLC and SDS-PAGE (Fig 9).

## UDA binding to RBD

To further determine whether UDA can inhibit RBD, the ELISA method was used for the study. The differences in the optical density of cells with or without the UDA treatment with

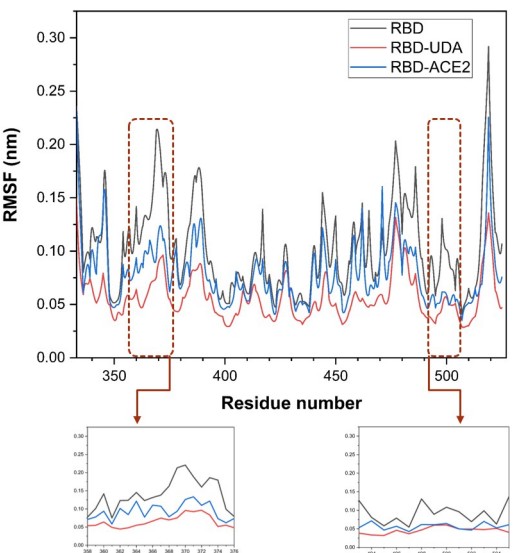

**Fig 6. Comparison of Cα atom RMSF of free RBD and RBD bonded UDA in 10 final nanoseconds of molecular dynamics simulation.**

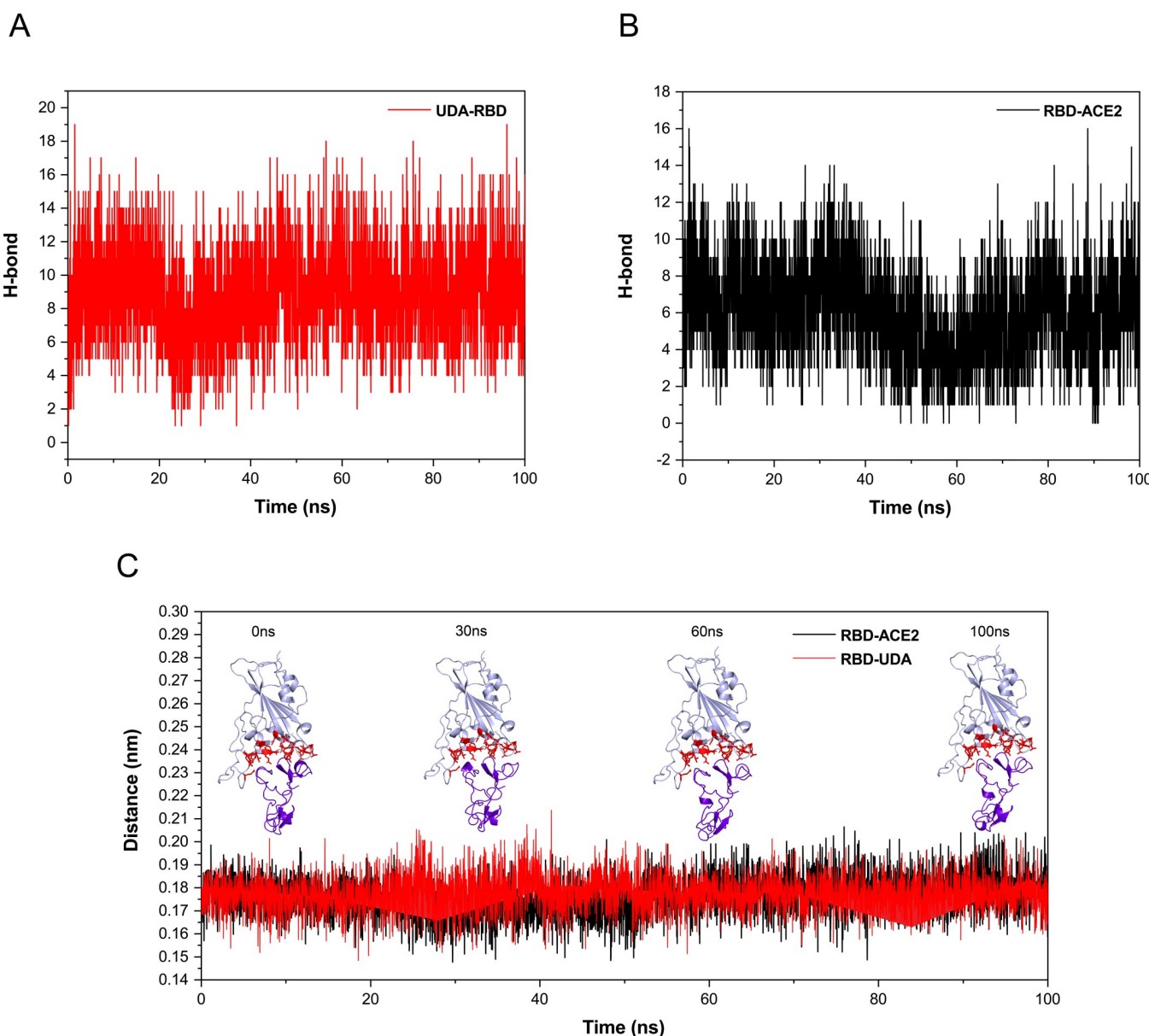

**Fig 7.** The plots of intermolecular hydrogen bonds between active sites of RBD-UDA (a) and RBD-ACE2 during 100 ns of molecular dynamic simulations (b). The change of distance between centers of active sites UDA and RBD (red) and ACE2 and RBD (black) over the period of 100 ns (c).

different dilutions showed that UDA was significantly bound to RBD. We compared the mean absorbance of Cells treated with serum IgG of 3 patients (21 days post-infection) as a positive control with the absorbance of Cells treated with UDA. The obtained results show that all absorbances at 450 nm are close to 420 (Fig 10).

## UDA toxicity

The results of the MTT assay are summarized in Fig 11, which shows that the percentage of cell viability depends on the UDA dilutions. The toxicity is 35–45% in 1:1 and 20–30% in 1:729 dilutions. These results suggest that UDA may be safe in human cells.

**Table 3. Hydrogen bond occupancy between RBD and ACE2, RBD and UDA during 100ns molecular dynamic simulations.**

| | RBD-ACE2 | | | RBD-UDA | |
| --- | --- | --- | --- | --- | --- |
| **Donor** | **Acceptor** | **Occupancy (%)** | **Donor** | **Acceptor** | **Occupancy (%)** |
| TYR505 (HH) | GLU37 (OE2) | 84.2 | TYR76 (HH) | TYR505 (OH) | 66.9 |
| GLY502 (H) | LYS353 (O) | 98.6 | GLY502 (H) | ASP75 (OD2) | 23.8 |
| ASN501 (D21) | TYR41 (OH) | 10.1 | GLY502 (H) | ASP75 (OD1) | 39.4 |
| THR500 (HG1) | ASP355 (OD1) | 11.5 | ASN501 (D21) | ASN81 (OD1) | 33.5 |
| THR500 (HG1) | TYR41 (OH) | 84.4 | ASN501 (D21) | ASP75 (OD1) | 30.1 |
| LYS353 (HZ1) | GLY496 (O) | 13.1 | ASN501 (D21) | ASP75 (OD2) | 14.2 |
| GLN493 (E21) | HIS34 (O) | 52.4 | HIS67 (HE2) | ASN501 (OD1) | 92.3 |
| TYR88 (HH) | ASN487 (OD1) | 97.2 | THR500 (HG1) | ASN81 (OD1) | 19.4 |
| ASN487 (D21) | GLN24 (OE1) | 26.5 | THR500 (HG1) | GLY79 (O) | 38.9 |
| SER19 (H1) | SER477 (OG) | 45.0 | ASN81 (D21) | THR500 (O) | 53.5 |
| SER19 (HG) | ALA475 (O) | 53.2 | ASN81 (D21) | THR500 (OG1) | 21.3 |
| HIS34 (HD1) | TYR453 (OH) | 34.8 | GLN498 (E21) | ASN81 (OD1) | 37.3 |
| LYS417 (HZ1) | HIS34 (NE2) | 28.1 | GLN498 (E21) | GLY80 (O) | 81.8 |
| LYS417 (HZ1) | ASP30 (OD2) | 45.3 | GLY496 (H) | HIS67 (ND1) | 88.4 |
| LYS417 (HZ1) | ASP30 (OD1) | 66.7 | GLN493 (E21) | HIS67 (O) | 19.0 |
| | | | GLN493 (E21) | CYS49 (O) | 13.7 |
| | | | PHE490 (H) | ALA52 (O) | 34.2 |
| | | | ARG48 (H21) | TYR453 (OH) | 39.9 |
| | | | ARG48 (HE) | TYR453 (OH) | 55.0 |
| | | | GLN83 (E21) | TYR449 (OH) | 16.4 |

## Discussion

Due to the limitations of regular updates of vaccines, such as a long time for the clinical trial, efficacy, and side effects, UDA can be used as an alternative treatment like some antiviral drugs used during an outbreak (9). Except that, UDA can be helpful because its production time is much shorter and its source is herbal, and it's found worldwide (Europe, Africa, Asia, America) [28]. Komaki et al. (14) reported that the UDA inhibits SARS-CoV from attaching to

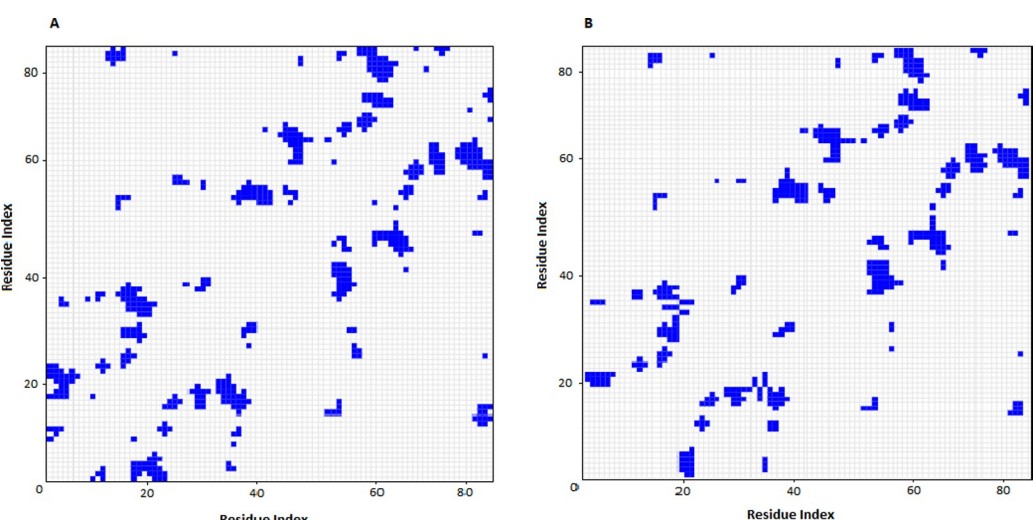

**Fig 8.** Protein contact map between UDA and RBD at 0 ns (a) and 100 ns (b) of molecular dynamic simulation.

**Table 4. Binding free energies from MM-PBSA calculation.**

| Type of Energeis | (kJ/mol) |
|---|---|
| ΔE van der Waals | -333.523 +/- 29.347 |
| ΔE electrostatic | -187.920 +/- 32.598 |
| ΔE Polar solvation | 508.793 +/- 108.926 |
| ΔE SASA | -34.855 +/- 2.174 |
| ΔG total (Binding energy) | -47.505 +/- 12.401 |

cells and is probably due to the binding to the N-acetyl glucosamine-like residues present on the glycoprotein spike. As expected, In other studies this carbohydrate-binding to SARS-CoV-2 spike glycoprotein was confirmed [17, 29]. However, we examined the binding without considering the protein surface glycans. Initially, UDA docking was performed with five target proteins (Tmprss2, plpro, 3clmainpro, ACE2, and RBD); UDA did not bind significantly to other target proteins except RBD. Residues of RBD, Tyr505, Gly502, Gly496, Asn501, Thr500, Gln498, Tyr449, Gly446, Tyr489, Phe456, Lys417, Gln493, Leu455, and Tyr453 involved in interaction with UDA are also present in connection with ACE2. The binding energy of RBD-UDA, -12.2 kcal/mol, and the binding energy of RBD-ACE2, -12.4 kcal/mol, has been calculated, which are close to each other. The 100 ns molecular dynamic simulation in an aqueous environment showed rearrangement of the position of the UDA and RBD complex. It was found that RBD in the RBD-UDA complex has the lowest RMSD value compared to the other two modes (free RBD and RBD in the RBD-ACE2 complex) (Fig 4). Also, the most significant difference in the mobility of RMSF plots of free RBD compared to RMSF plots of RBD-UDA and RBD-ACE2 in the regions of residues 358–376 and 493–505, including the binding site to ACE2. The fluctuations in these two regions are significantly reduced in the RBD-UDA, as expected, the reason for the decrease in region 493–505 is due to the formation

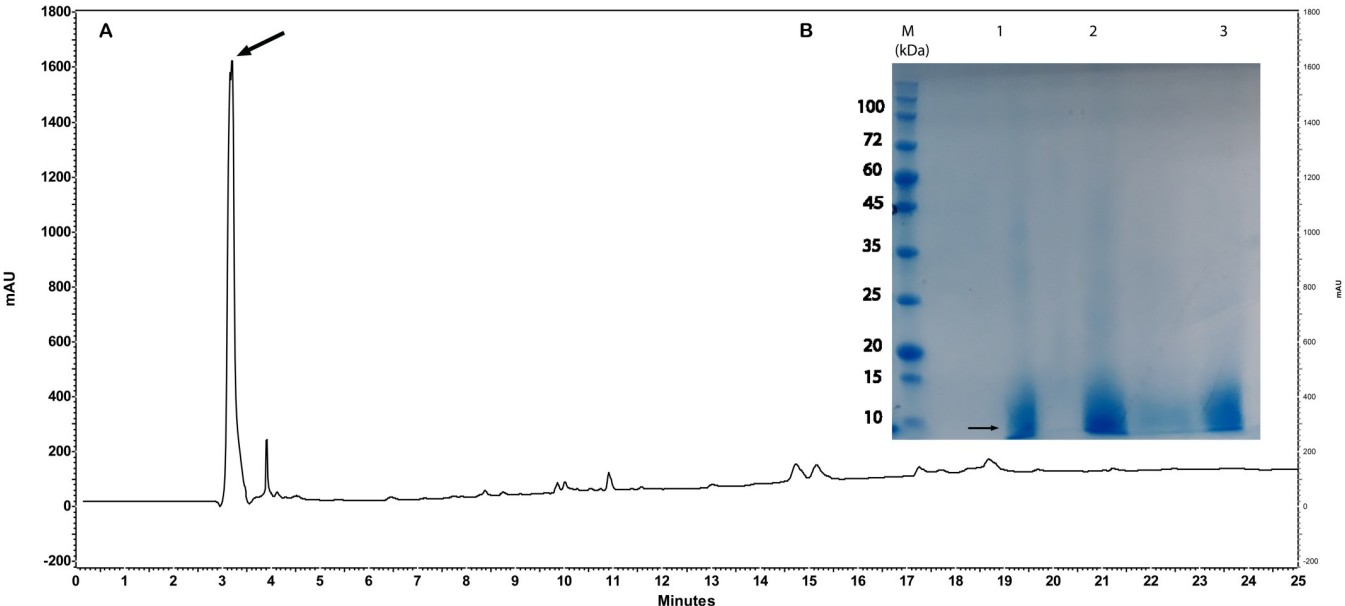

**Fig 9. Purity analysis of UDA peptide. a** RP-HPLC analysis of purified UDA using a C18 column. 150 µg/ml of purified UDA, at a flow rate of 1 ml/min was injected on column. The effluent was monitored by recording UV absorbance at 214 nm. Absorbance is in milliabsorbance units (mAU). **b** SDS-PAGE analysis of UDA as well as compliance with the purification procedures (The lanes 1 to 3). Lane M indicates a molecular-weight size marker.

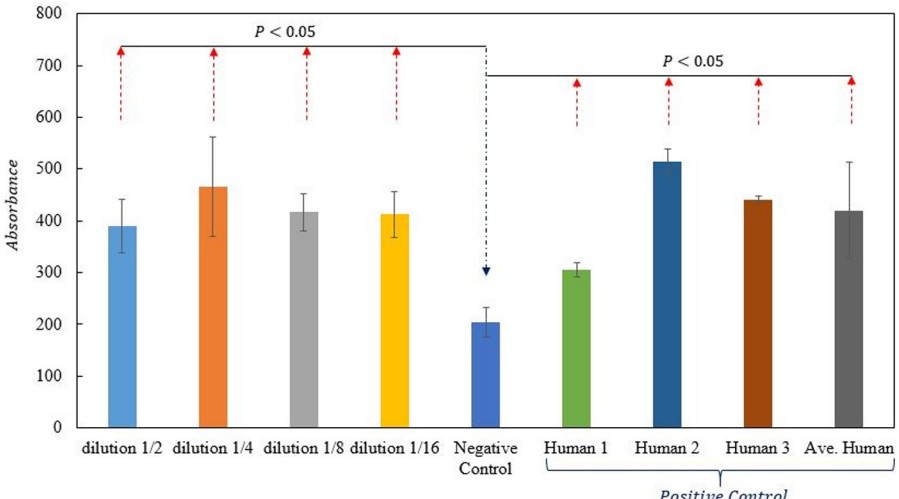

**Fig 10. Comparative ELISA for three human patient serums with IgG subclass and UDA-treated plates.** ELISA results from four independent experiments of binding of UDA to the RBD of SARS-CoV-2 spike protein with different dilutions, negative control, three samples of convalescent serum IgG from patients, average of results of patient samples as positive control.

of 10 hydrogen bonds by Gln493, Ser494, Tyr495, Gly496, Phe497, Gln498, Thr500, Asn501, Gly502 and Tyr505 in this region of RBD with UDA. In the study of Sayan Batacharaji et al. [30] comparison of RMSF plots of free RBD and ACE2-bound RBD illustrate that ACE2--bound RBD plot decreases simultaneously in the dynamics of loop 474–486 and loop 358–376, possibly due to allosteric involvement of loop 358–376 through correlated motion. These observations also occurred in both RBD-UDA and RBD-ACE2 simulations in our study. It is generally observed that the fluctuations of the RBD-UDA plot are less than other plots (Fig 6).

After 100ns of simulation, the RBD residues in binding formation are Phe456, Tyr489, Gln493, Gly502, Asn501, Tyr505, Tyr453, Lys417, Gln498, and Thr500, which it shows the RBD-UDA complex is stable in 100 ns. From the distance plot, it was observed that RBD does

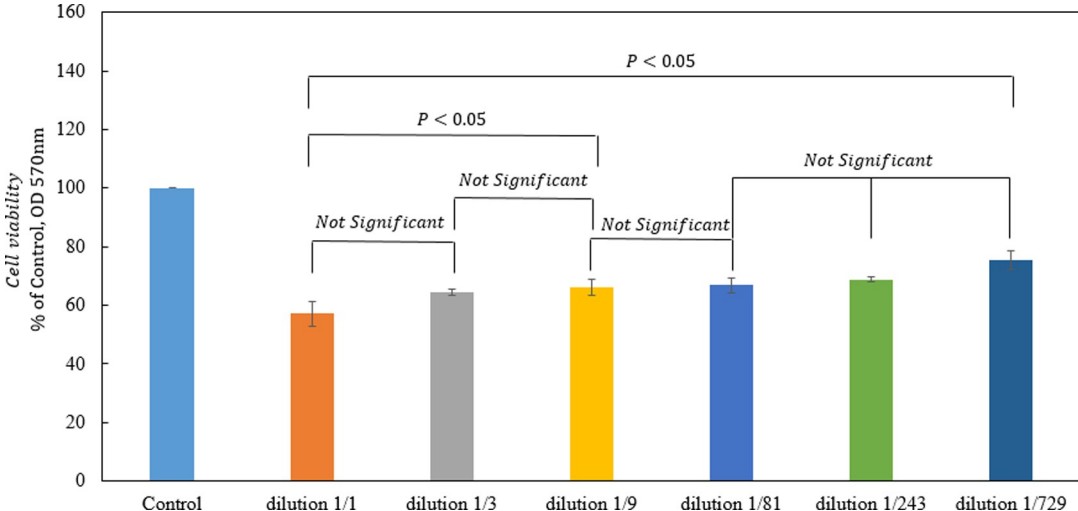

**Fig 11. Three independent determinations of cellular viability by MTT assay.** Comparison of percentage of viability of cells exposed to 1:3, 1:9, 1:27, 1:81, 1:243, and 1:729 dilutions of UDA.

not separate from ACE2 during the simulation, it also keeps its distance from UDA (average 1.8 Å), so in 100 ns of the simulation, UDA does not move away from RBD. The RBD-UDA hydrogen bond pattern showed an average of 10 hydrogen bonds formed in 100 ns and up to eighteen hydrogen bonds at some points. Also, a comparison of residual contact maps at 0 ns and 100 ns showed that most connected residues had retained their connection. Thus, the RBD-UDA complex was stable in an aqueous medium. The energy parameters calculation was performed by MM-PBSA so that snapshots were extracted every 1 ns at regular intervals from the 100 nanoseconds of simulation. The average binding energy was calculated to be -47.505kJ/mol. To confirm our in silico studies, the ELISA assay was used. The obtained results show a significant difference between absorbance of UDA-treated cells and negative control, and its closeness to positive control is a sign of UDA-RBD conjunction.

In summary, UDA can potentially inhibit the RBD of SRAS-CoV-2 and host cell receptor interaction. However, in vivo and ex vivo trials are necessary for further confirmation of this conclusion. We want to share our findings with antivirus researchers at the first opportunity; the results suggest that UDA can be a potential treatment for SARS-CoV-2 infection, and the RBD-UDA complex may be a novel type of vaccine for future clinical trials of SARS-CoV-2 [31].

## Supporting information

**S1 File.**
(PDB)

## Author Contributions

**Formal analysis:** Fatemeh Sabzian-Molaei.

**Investigation:** Fatemeh Sabzian-Molaei, Mohammad Sabzian-Molaei.

**Methodology:** Alireza Fattah Pour, Ahmad Molaei Rad.

**Project administration:** Mohammad Ali Nasiri Khalili, Amin Hadi.

**Software:** Amin Hadi.

**Supervision:** Mohammad Ali Nasiri Khalili, Hosein Shahsavarani, Amin Hadi.

**Validation:** Fatemeh Sabzian-Molaei, Hosein Shahsavarani, Amin Hadi.

**Writing – original draft:** Fatemeh Sabzian-Molaei.

**Writing – review & editing:** Amin Hadi.

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
