## [Decision Letter · Decision Letter 0]

11 Feb 2022

PONE-D-22-01270Urtica dioica Agglutinin a plant protein candidate for inhibition of SARS-COV-2 receptor-binding domain for control of Covid19 Infection: Molecular Dynamics Simulation and Experimental ApproachPLOS ONE

Dear Dr. Nasiri Khalili,

Thank you for submitting your manuscript to PLOS ONE. After careful consideration, we feel that it has merit but does not fully meet PLOS ONE’s publication criteria as it currently stands. Therefore, we invite you to submit a revised version of the manuscript that addresses the points raised during the review process.

ACADEMIC EDITOR: Please see the detailed comments below.

We look forward to receiving your revised manuscript.

Kind regards,

Mohd Adnan, PhD

Academic Editor

PLOS ONE

Journal Requirements:

2.In your Data Availability statement, you have not specified where the minimal data set underlying the results described in your manuscript can be found. PLOS defines a study's minimal data set as the underlying data used to reach the conclusions drawn in the manuscript and any additional data required to replicate the reported study findings in their entirety. All PLOS journals require that the minimal data set be made fully available. For more information about our data policy, please see http://journals.plos.org/plosone/s/data-availability.

Additional Editor Comments:

Reviewers have commented against the acceptance of manuscript in its current form. Manuscript suffers from serious concerns regarding the implemented protocol as well as presentation of the data. Please revise in light of reviewers comments and resubmit accordingly.

Reviewers' comments:

Reviewer's Responses to Questions

**Comments to the Author**

1. Is the manuscript technically sound, and do the data support the conclusions?

Reviewer #1: Partly

Reviewer #2: Partly

2. Has the statistical analysis been performed appropriately and rigorously? 

Reviewer #1: No

Reviewer #2: N/A

3. Have the authors made all data underlying the findings in their manuscript fully available?

Reviewer #1: No

Reviewer #2: Yes

4. Is the manuscript presented in an intelligible fashion and written in standard English?

Reviewer #1: No

Reviewer #2: No

5. Review Comments to the Author

Reviewer #1: The authors used Urtica dioica Agglutinin a plant protein to dock with RBD and performed MD and other studies to validate it. Though the article sounds interesting, following are my concerns

Major

1. The authors have not clearly mentioned the active site residues of RBD. They should list it in a table. Also what parameters were used in Cluspro?? they should mention it.

2. LigPlot of RBD-ACE2 crystal structure and RBD-UDA should be provided for comparison. Figure 2 contains only one ligplot (RBD-UDA). Quality of ligplot appears poor. Provide good resolution figure, to compare

3. Authors have not clearly mentioned RMSD of what is computed and discussed. Is it backbone, or c-alpha or entire protein complex or only RBD RMSD?? Authors should provide RMSD ( backbone ) of RBD alone, RBD in RBD-UDA, RBD in RBD-ACE2, UDA alone, and ACE-2 alone. Without which stability of the system cannot be discussed logically.

4. How is H-bond computed from trajectory? Authors should create index file of active site residues first and then should fed it to gromacs command to compute h-bond between activesite of RBD and ACE-2 and UDA. This will allow one to know whether h-bond formation is with active site or any other part of the protein.

5. Also, h-bond occupancy (from 100ns trajectory) in % for each active site residue should be computed and should be listed as table.

6. The distance plot is meaningless, unless author mention that it is the distance between what and what! Authors mentioned distance between RBD and UDA and RBD and ACE2. Authors should clearly mention is it between center of two proteins, or center of activesite. Distance can be computed using two points, authors should clearly mention what those two points are, it cannot be the whole proteins.

7. Radius of gyration and SASA plots should be provided for both the complexes. Also the results should be discussed.

8. why mmgbsa energies are computed only from last 20ns of the simulation??? It is biased to check energies only on last 20ns, authors should parse the trajectory for every 1ns (0-100ns, which contains only 100 frames) and should compute mmgbsa energies. Those should be plotted or provided as table.

9. Authors should perform their wet lab experiments in triplicates and should report the results. Current procedure does not include any statistical measures to test the significance. Authors should use statistical method to validate the wetlab results.

Minor

1. A lot of typo errors (especially spaces between words)

2. Language should be improved for readability.

3. Try to provide high resolution figures.

Reviewer #2: Reviewer Comments: Appears to use a lot of tools and has drawn conclusions from them but the details on the parameters are missing and information need to be clarified on the approach and analysis. While there are clear results, its difficult to accept some of them as these details are missing or unclear. Additional analysis of the computational data would aid in the statements on the stability of the docked and simulated structures. More detailed comments below.

Minor Revisions:

Missing citations for several methodologies/approaches mentioned. Ie GROMACS, MTT, PRODIGY etc

Supporting information with the HPLC data and other additional analysis that is not shown but noted (visualizations) would also help.

There are some grammatical errors that also need to be addressed. Typos are notable in the document particularly in the introduction, methods and discussion

Introduction

• Ending sentence is a leading statement of an incomplete thought. Consider revising “However, additional experimental studies are needed to confirm this capability.”

Methods

• Unclear, possibly used the word "supported" instead of "supplied". Better phrasing is needed “SDS-PAGE and High performance liquid chromatography (HPLC) analyzes verified it (data not shown). The recombinant receptor-binding domain (RBD) of SARS-CoV2 was supported by Pasteur Institute (Tehran, Iran). The human embryonic kidney cell line (HEK 293) was supported by the cell bank of Pasteur Institute (Tehran, Iran)”

• Automatic or auto “An automat docking tool” not sure of meaning

Discussion

• The first paragraph of the discussion is not needed and just repeats the introduction. First sentence of the 2nd paragraph of the discussion needs work and is an incomplete thought.

• Second to last paragraph “These observations also occurred in both RBD-UDA and RBD-UDA simulations in our study.”

Questions and Major Revisions:

1. The calculation of the mm-pbsa was done for the last 20ns. How was that time determined and why do you consider it sufficient?

2. Hbond and Distance are blurry and hard to read. A spline or trendline would help. Also were you using the hydrogen bonds with the default settings in GROMACS or did you specify the distance and angle?

3. The MD simulation was in a solvent environment but the box type, salt concentration or number of ions are not listed. These details are needed.

4. The authors also say little in the analysis of the MD simulation. What happened between 20-40ns in the UDA-RBD simulation. Your hydrogen bonds dropped and distance increased.

5. The docking approach is missing details on the approach. Beyond the binding energy how did you quantify the “best docked system”?

6. The RMSD was compared between the UDA-RBD and ACE-RBD complexes the complexes are different and not directly comparable. The RBD in both cases is comparable but the way its written indicates you are comparing the complexes not the RBD in those complexes. Please clarify or justify this analysis

7. No starting structures or simulation snapshots were shown in the document. Was the starting structure for both complexes in your simulation of the docked configurations and was that the most stable conformation according to the simulation? Your data indicates that there had to be some changes in the orientation. How did the final simulation configuration compare to you other docked complexes and the selected one? What happened visually and with the complex in the 20-40ns RBD-UDA simulation?

8. The docking results are for a single conformation and didn't mention any other conformations or orientations. What were the result and scoring for the other conformations/orientations? Were they significantly different? You state you chose the best docking conformation to use for comparison and MD studies. To do so you have to have some comparison with other conformations/orientations. This needs to be highlighted and explained. Also if these conformations were close, how did they compare to the MD simulation snapshots?

9. The authors mention the residues for the RBD that are interacting but not the binding pairs formed with UDA and those resides. Are the binding pairs (specific UDA residues with specific RBD residues) consistent? A calculation of the percent binding during the simulation time would have helped. This would show that the binding pairs shown in the final conformation are those most stable throughout the simulation and lend to the stability of the complex.

6. PLOS authors have the option to publish the peer review history of their article (what does this mean?). If published, this will include your full peer review and any attached files.

Reviewer #1: No

Reviewer #2: No

---

## [Author Response · Author response to Decision Letter 0]

12 Mar 2022

Dear Editor-in-Chief of PLOS ONE

I cordially appreciate your reviewers for giving the best comments in order to upgrade our paper. I believe that our paper have significantly improved with these comments. Anyway, all the reviewers' comments have perfectly done which are responded as follows:

Best regards

Amin Hadi

Reviewer 1

Major

Query 1

The authors have not clearly mentioned the active site residues of RBD. They should list it in a table. Also what parameters were used in Cluspro?? they should mention it.

Response

This information added in Table 1.

Table 1. List of interacting residues between RBD and ACE2

Distance

(A°) 

ACE2

RBD

Interactions

2.81 SER 19 ALA 475 

2.87 GLN 24 ASN 487 

2.86 ASP 30 LYS 417 

2.83 ASP 38 TYR 449 

3.09 GLN 42 Tyr 449 

2.73 TYR 41 THR 500 

2.57 GLN 42 GLN 498 

2.82 TYR 83 ASN 487 

2.79 LYS 353 GLY 502 

3.22 GLY 496 LYS 353 

2.08 GLU 37 TYR 505 

2.86 

ASP 30 

LYS 417

The docking modes generated from Cluspro were visualized in PyMOL (19). The selection of the best RBD-UDA complex was based on two factors: The first factor was the lowest energy using the balanced method based on the PIPER docking algorithm, which is calculated based on the following expression the interaction energy between two proteins: E = w1Erep + w2Eattr + w3Eelec + w4EDARS, here Erep and Eattr show the repulsive and attractive contributions to the van der Waals interaction energy. Eelec is the electrostatics energy and EDARS is a pairwise structure-based potential constructed by the Decoys as the Reference State (DARS) approach. Coefficients within the equation w1, w2, w3, and w4 define the weights of the terms(19)(21). The second choosing factor was more attachment to the active site of RBD. Finally, The best complex was selected and analyzed using PDBSUM and LigPlot+ (21). The binding free energy of the best-docked model was calculated in PRODIGY, an online protein binding energy prediction server for ΔG affinity binding (kcal/mol) and complex stability. PRODIGY is a set of web-based services available at (https://bianca.science.uu.nl/prodigy/). PRODIGY focuses on predicting binding affinity within biologic complexes (22).

Query 2

LigPlot of RBD-ACE2 crystal structure and RBD-UDA should be provided for comparison. Figure 2 contains only one ligplot (RBD-UDA). Quality of ligplot appears poor. Provide good resolution figure, to compare

Response

The required items were added in Figure 3 and the quality of this figure was improved.

Query 3

Authors have not clearly mentioned RMSD of what is computed and discussed. Is it backbone, or c-alpha or entire protein complex or only RBD RMSD?? Authors should provide RMSD ( backbone ) of RBD alone, RBD in RBD-UDA, RBD in RBD-ACE2, UDA alone, and ACE-2 alone. Without which stability of the system cannot be discussed logically.

Response

This item has been done.

Figure 4. RMSD (backbone) plots for free RBD, and RBD in the RBD-UDA, and RBD-ACE2 complexes (a). RMSD (backbone) of free UDA and UDA in the RBD-UDA complex during 100 ns of molecular dynamics simulation (b).

Query 4

How is H-bond computed from trajectory? Authors should create index file of active site residues first and then should fed it to gromacs command to compute h-bond between activesite of RBD and ACE-2 and UDA. This will allow one to know whether h-bond formation is with active site or any other part of the protein.

Response

This item has been done (Figure 7).

Figure 7. The plots of intermolecular hydrogen bonds between active sites of RBD-UDA (a) and RBD-ACE2 during 100 ns of molecular dynamic simulations (b). The change of distance between centers of active sites UDA and RBD (red) and ACE2 and RBD (black) over the period of 100 ns (c).

Query 5

Also, h-bond occupancy (from 100ns trajectory) in % for each active site residue should be computed and should be listed as table.

Response

Table 3. Hydrogen bond occupancy between RBD and ACE2, RBD and UDA during 100ns molecular dynamic simulations.

 RBD-UDA RBD-ACE2 

Occupancy (%) Acceptor Donor Occupancy (%) Acceptor Donor

66.9 TYR505 (OH ) TYR76 (HH ) 84.2 GLU37 (OE2) TYR505 (HH )

23.8 ASP75 (OD2) GLY502 (H) 98.6 LYS353 (O) GLY502 (H)

39.4 ASP75 (OD1) GLY502 (H) 10.1 TYR41 (OH ) ASN501 (D21)

33.5 ASN81 (OD1) ASN501 (D21) 11.5 ASP355 (OD1) THR500 (HG1)

30.1 ASP75 (OD1) ASN501 (D21) 84.4 TYR41 (OH) THR500 (HG1)

14.2 ASP75 (OD2) ASN501 (D21) 13.1 GLY496 (O) LYS353 (HZ1)

92.3 ASN501 (OD1) HIS67 (HE2) 52.4 HIS34 (O) GLN493 (E21)

19.4 ASN81 (OD1) THR500 (HG1) 97.2 ASN487 (OD1) TYR88 (HH )

38.9 GLY79 (O) THR500 (HG1) 26.5 GLN24 (OE1) ASN487 (D21)

53.5 THR500 (O) ASN81 (D21) 45.0 SER477 (OG ) SER19 (H1)

21.3 THR500 (OG1) ASN81 (D21) 53.2 ALA475 (O) SER19 (HG )

37.3 ASN81 (OD1) GLN498 (E21) 34.8 TYR453 (OH ) HIS34 (HD1)

81.8 GLY80 (O) GLN498 (E21) 28.1 HIS34 (NE2) LYS417 (HZ1)

88.4 HIS67 (ND1) GLY496 (H) 45.3 ASP30 (OD2) LYS417 (HZ1)

19.0 HIS67 (O) GLN493 (E21) 66.7 ASP30 (OD1) LYS417 (HZ1)

13.7 CYS49 (O) GLN493 (E21) 

34.2 ALA52 (O) PHE490 (H) 

39.9 TYR453 (OH) ARG48 (H21) 

55.0 TYR453 (OH) ARG48 (HE) 

16.4 TYR449 (OH) GLN83 (E21) 

Query 6

The distance plot is meaningless, unless author mention that it is the distance between what and what! Authors mentioned distance between RBD and UDA and RBD and ACE2. Authors should clearly mention is it between center of two proteins, or center of activesite. Distance can be computed using two points, authors should clearly mention what those two points are, it cannot be the whole proteins.

Response

The distance between centers of active sites of RBD and UDA and centers of active sites of RBD and ACE2 was analyzed using GROMACS software, and the results showed that UDA does not separate from RBD like ACE2. The snapshots taken at different times of the simulation confirm its stable connection (Figure7.c).

Figure 7. The plots of intermolecular hydrogen bonds between active sites of RBD-UDA (a) and RBD-ACE2 during 100 ns of molecular dynamic simulations (b). The change of distance between centers of active sites UDA and RBD (red) and ACE2 and RBD (black) over the period of 100 ns (c).

Query 7

Radius of gyration and SASA plots should be provided for both the complexes. Also the results should be discussed.

Response

The radius of gyration (Rg), Solvent-accessible surface area (SASA), Root mean square fluctuation (RMSF) of the α-carbons of systems during the simulations, intermolecular hydrogen bond profile of active site of complexes, distance fluctuation plot (mindist), and comparison of residual contact patterns between UDA peptide and RBD of S protein at 0 ns and 100 ns were studied, contact maps were visualized using CMWeb (23).

The Rg and SASA plots for RBD in the RBD-UDA and RBD-ACE2 complexes show stable fluctuations over the time of the simulations (Figure5). The slight decrease in the Rg and SASA values of RBD in the RBD-UDA complex indicates that binding to UDA may increase the overall compactness of RBD.

Figure 5. Plots of Rg (a) and SASA (b) for RBD in RBD-UDA and RBD-ACE2 complexes.

Query 8

why mmgbsa energies are computed only from last 20ns of the simulation??? It is biased to check energies only on last 20ns, authors should parse the trajectory for every 1ns (0-100ns, which contains only 100 frames) and should compute mmgbsa energies. Those should be plotted or provided as table.

Response

The reason for choosing the last 20 nanoseconds is that the system is at its most stable. However, it was recalculated for 100 nanoseconds as recommended by the reviewer.

Table 4. Binding free energies from MM-PBSA calculation.

Type of Energeis (kJ/mol)

ΔE van der Waals -333.523

ΔE electrostatic -187.920

ΔE Polar solvation 508.793

ΔE SASA -34.855

ΔG total (Binding energy) -47.505

Query 9

Authors should perform their wet lab experiments in triplicates and should report the results. Current procedure does not include any statistical measures to test the significance. Authors should use statistical method to validate the wetlab results.

Response

Minor

Query 1

A lot of typo errors (especially spaces between words)

Response

Some errors and sentences were corrected according to your valuable comment.

Query 2

Language should be improved for readability.

Response

The text is fully checked and all typos and grammatical errors have been corrected.

Query 3

Try to provide high resolution figures.

Response

This item has been done.

Thank you for your comments and suggestions.

Reviewer 2

Major

Query 1

The calculation of the mm-pbsa was done for the last 20ns. How was that time determined and why do you consider it sufficient?

Response

The reason for choosing the last 20 nanoseconds is that the system is at its most stable. However, it was recalculated for 100 nanoseconds as recommended by the reviewer.

Table 4. Binding free energies from MM-PBSA calculation.

Type of Energeis (kJ/mol)

ΔE van der Waals -333.523

ΔE electrostatic -187.920

ΔE Polar solvation 508.793

ΔE SASA -34.855

ΔG total (Binding energy) -47.505

Query 2

Hbond and Distance are blurry and hard to read. A spline or trendline would help. Also were you using the hydrogen bonds with the default settings in GROMACS or did you specify the distance and angle?

Response

The number of H-bonds between active sites of RBD and ACE-2 and UDA, calculated using the default settings of the g-h bond tool in Gromacs and the quality of this figure was improved.

Query 3

The MD simulation was in a solvent environment but the box type, salt concentration or number of ions are not listed. These details are needed.

Response

Molecular dynamics for 100 ns were done using the GROMACS 2020.2 software program: SPC216 water model in a cubic box and amber99sb.ildn force field. Adding a suitable number of CL- /Na+ ions neutralized the system. Therefore, for free RBD, RBD-UDA, RBD-ACE2, and Free UDA simulations, 2CL-, 3CL-, 24Na+, and 1CL- were added, respectively.

Query 4

The authors also say little in the analysis of the MD simulation. What happened between 20-40ns in the UDA-RBD simulation. Your hydrogen bonds dropped and distance increased.

Response

After evaluating the hydrogen bonds and the distance from the active sites, this decrease and increase is not very noticeable. Also, in the snapshots obtained during the simulation, shown in Figure 7, the connection was not disconnected and did not change direction.

Figure 7. The plots of intermolecular hydrogen bonds between active sites of RBD-UDA (a) and RBD-ACE2 during 100 ns of molecular dynamic simulations (b). The change of distance between centers of active sites UDA and RBD (red) and ACE2 and RBD (black) over the period of 100 ns (c).

Query 5

The docking approach is missing details on the approach. Beyond the binding energy how did you quantify the “best docked system”?

Response

The selection of the best RBD-UDA complex was based on two factors: The first factor was the lowest energy using the balanced method based on the PIPER docking algorithm, which is calculated based on the following expression the interaction energy between two proteins: E = w1Erep + w2Eattr + w3Eelec + w4EDARS, here Erep and Eattr show the repulsive and attractive contributions to the van der Waals interaction energy. Eelec is the electrostatics energy and EDARS is a pairwise structure-based potential constructed by the Decoys as the Reference State (DARS) approach. Coefficients within the equation w1, w2, w3, and w4 define the weights of the terms(19)(21). The second choosing factor was more attachment to the active site of RBD. The best RBD complex with UDA was selected and analyzed using Molegro and LigPlot+ Finally, The best complex was selected and analyzed using PDBSUM and LigPlot+.

Query 6

The RMSD was compared between the UDA-RBD and ACE-RBD complexes the complexes are different and not directly comparable. The RBD in both cases is comparable but the way its written indicates you are comparing the complexes not the RBD in those complexes. Please clarify or justify this analysis.

Response

This item clarified in text.

Query 7

No starting structures or simulation snapshots were shown in the document. Was the starting structure for both complexes in your simulation of the docked configurations and was that the most stable conformation according to the simulation? Your data indicates that there had to be some changes in the orientation. How did the final simulation configuration compare to you other docked complexes and the selected one? What happened visually and with the complex in the 20-40ns RBD-UDA simulation?

Response

After evaluating the hydrogen bonds and the distance from the active sites, this decrease and increase is not very noticeable. Also, in the snapshots obtained during the simulation, shown in Figure 7, the connection was not disconnected and did not change direction.

Figure 7. The plots of intermolecular hydrogen bonds between active sites of RBD-UDA (a) and RBD-ACE2 during 100 ns of molecular dynamic simulations (b). The change of distance between centers of active sites UDA and RBD (red) and ACE2 and RBD (black) over the period of 100 ns (c).

Query 8

The docking results are for a single conformation and didn't mention any other conformations or orientations. What were the result and scoring for the other conformations/orientations? Were they significantly different? You state you chose the best docking conformation to use for comparison and MD studies. To do so you have to have some comparison with other conformations/orientations. This needs to be highlighted and explained. Also if these conformations were close, how did they compare to the MD simulation snapshots?

Response

According to RBD active site and Cluspro scoring, the selected model, among the top Five Cluspro compounds, had the highest involvement with the active site, with a significant difference compared to other models.

Figure . The top five docked complexes for UDA-RBD

Query 9

The authors mention the residues for the RBD that are interacting but not the binding pairs formed with UDA and those resides. Are the binding pairs (specific UDA residues with specific RBD residues) consistent? A calculation of the percent binding during the simulation time would have helped. This would show that the binding pairs shown in the final conformation are those most stable throughout the simulation and lend to the stability of the complex.

Response

On the other hand, the Occupancy of the hydrogen bonds between UDA and ACE during the simulation shows in the table3, which confirms that the UDA maintains its hydrogen bonds with active site residues of RBD such as TYR505, GLY502, ASN501, THR500, GLN498, GLY496, PHE490, TYR453 and TYR449 during simulation.

Table 3. Hydrogen bond occupancy between RBD and ACE2, RBD and UDA during 100ns molecular dynamic simulations.

 RBD-UDA RBD-ACE2 

Occupancy (%) Acceptor Donor Occupancy (%) Acceptor Donor

66.9 TYR505 (OH ) TYR76 (HH ) 84.2 GLU37 (OE2) TYR505 (HH )

23.8 ASP75 (OD2) GLY502 (H) 98.6 LYS353 (O) GLY502 (H)

39.4 ASP75 (OD1) GLY502 (H) 10.1 TYR41 (OH ) ASN501 (D21)

33.5 ASN81 (OD1) ASN501 (D21) 11.5 ASP355 (OD1) THR500 (HG1)

30.1 ASP75 (OD1) ASN501 (D21) 84.4 TYR41 (OH) THR500 (HG1)

14.2 ASP75 (OD2) ASN501 (D21) 13.1 GLY496 (O) LYS353 (HZ1)

92.3 ASN501 (OD1) HIS67 (HE2) 52.4 HIS34 (O) GLN493 (E21)

19.4 ASN81 (OD1) THR500 (HG1) 97.2 ASN487 (OD1) TYR88 (HH )

38.9 GLY79 (O) THR500 (HG1) 26.5 GLN24 (OE1) ASN487 (D21)

53.5 THR500 (O) ASN81 (D21) 45.0 SER477 (OG ) SER19 (H1)

21.3 THR500 (OG1) ASN81 (D21) 53.2 ALA475 (O) SER19 (HG )

37.3 ASN81 (OD1) GLN498 (E21) 34.8 TYR453 (OH ) HIS34 (HD1)

81.8 GLY80 (O) GLN498 (E21) 28.1 HIS34 (NE2) LYS417 (HZ1)

88.4 HIS67 (ND1) GLY496 (H) 45.3 ASP30 (OD2) LYS417 (HZ1)

19.0 HIS67 (O) GLN493 (E21) 66.7 ASP30 (OD1) LYS417 (HZ1)

13.7 CYS49 (O) GLN493 (E21) 

34.2 ALA52 (O) PHE490 (H) 

39.9 TYR453 (OH) ARG48 (H21) 

55.0 TYR453 (OH) ARG48 (HE) 

16.4 TYR449 (OH) GLN83 (E21) 

Minor

Query 1

Missing citations for several methodologies/approaches mentioned. Ie GROMACS, MTT, PRODIGY etc

Supporting information with the HPLC data and other additional analysis that is not shown but noted (visualizations) would also help.

There are some grammatical errors that also need to be addressed. Typos are notable in the document particularly in the introduction, methods and discussion

Introduction

• Ending sentence is a leading statement of an incomplete thought. Consider revising “However, additional experimental studies are needed to confirm this capability.”

Methods

• Unclear, possibly used the word "supported" instead of "supplied". Better phrasing is needed “SDS-PAGE and High performance liquid chromatography (HPLC) analyzes verified it (data not shown). The recombinant receptor-binding domain (RBD) of SARS-CoV2 was supported by Pasteur Institute (Tehran, Iran). The human embryonic kidney cell line (HEK 293) was supported by the cell bank of Pasteur Institute (Tehran, Iran)”

• Automatic or auto “An automat docking tool” not sure of meaning

Discussion

• The first paragraph of the discussion is not needed and just repeats the introduction. First sentence of the 2nd paragraph of the discussion needs work and is an incomplete thought.

• Second to last paragraph “These observations also occurred in both RBD-UDA and RBD-UDA simulations in our study.”

Response

This item has been done.

Thank you for your comments and suggestions.

---

## [Decision Letter · Decision Letter 1]

18 Apr 2022

PONE-D-22-01270R1Urtica dioica Agglutinin a plant protein candidate for inhibition of SARS-COV-2 receptor-binding domain for control of Covid19 Infection: Molecular Dynamics Simulation and Experimental ApproachPLOS ONE

Dear Dr. Nasiri Khalili,

Thank you for submitting your manuscript to PLOS ONE. After careful consideration, we feel that it has merit but does not fully meet PLOS ONE’s publication criteria as it currently stands. Therefore, we invite you to submit a revised version of the manuscript that addresses the points raised during the review process.

We look forward to receiving your revised manuscript.

Kind regards,

Mohd Adnan, PhD

Academic Editor

PLOS ONE

Journal Requirements:

Additional Editor Comments (if provided):

Manuscript is significantly improved by the authors. However, there are still some minor concerns raised by the reviewer. Please address these concerns and resubmit accordingly.

Reviewers' comments:

Reviewer's Responses to Questions

**Comments to the Author**

1. If the authors have adequately addressed your comments raised in a previous round of review and you feel that this manuscript is now acceptable for publication, you may indicate that here to bypass the “Comments to the Author” section, enter your conflict of interest statement in the “Confidential to Editor” section, and submit your "Accept" recommendation.

Reviewer #1: All comments have been addressed

2. Is the manuscript technically sound, and do the data support the conclusions?

Reviewer #1: Yes

3. Has the statistical analysis been performed appropriately and rigorously? 

Reviewer #1: Yes

4. Have the authors made all data underlying the findings in their manuscript fully available?

Reviewer #1: No

5. Is the manuscript presented in an intelligible fashion and written in standard English?

Reviewer #1: Yes

6. Review Comments to the Author

Reviewer #1: The changes made were satisfactory. However, following changes could be made before acceptance.

1. Since the mmpbsa calculations were made from the entire MD trajectory, the Table 4 should include standard deviation (S.D +/-) values along with the energy values.

2. The authors may provide the best docked structure (PDB co-ordinates) available in the supplementary file. (not mandatory).

7. PLOS authors have the option to publish the peer review history of their article (what does this mean?). If published, this will include your full peer review and any attached files.

Reviewer #1: No

---

## [Author Response · Author response to Decision Letter 1]

19 Apr 2022

Dear Editor-in-Chief of PLOS ONE

I cordially appreciate your reviewers for giving the best comments in order to upgrade our paper. I believe that our paper have significantly improved with these comments. Anyway, all the reviewers' comments have perfectly done which are responded as follows:

Best regards

Amin Hadi

Reviewer 1

Minor

Query 1

Since the mmpbsa calculations were made from the entire MD trajectory, the Table 4 should include standard deviation (S.D +/-) values along with the energy values.

Response

This information is added in Table 4.

Table 4. Binding free energies from MM-PBSA calculation.

Type of Energeis (kJ/mol)

ΔE van der Waals -333.523 +/- 29.347

ΔE electrostatic -187.920 +/- 32.598

ΔE Polar solvation 508.793 +/- 108.926

ΔE SASA -34.855 +/- 2.174

ΔG total (Binding energy) -47.505 +/- 12.401

Query 2

The authors may provide the best docked structure (PDB co-ordinates) available in the supplementary file. (not mandatory)

Response

The best-docked structure (PDB format) is presented in the Supplementary Material.

Thank you for your comments and suggestions.

---

## [Editor Report · Decision Letter 2]

25 Apr 2022

Urtica dioica Agglutinin a plant protein candidate for inhibition of SARS-COV-2 receptor-binding domain for control of Covid19 Infection: Molecular Dynamics Simulation and Experimental Approach

PONE-D-22-01270R2

Dear Dr. Nasiri Khalili,

We’re pleased to inform you that your manuscript has been judged scientifically suitable for publication and will be formally accepted for publication once it meets all outstanding technical requirements.

Kind regards,

Mohd Adnan, PhD

Academic Editor

PLOS ONE

Additional Editor Comments (optional):

Manuscript is significantly improved by the authors and now can be accepted in its current form.